# Skotomorphogenesis exploits threonine to promote hypocotyl elongation

Hiromitsu Tabeta[1,2,3], Yasuhiro Higashi[2], Yozo Okazaki[2,4], Kiminori Toyooka[2] , Mayumi Wakazaki[2], Mayuko Sato[2], Kazuki Saito[2], Masami Y Hirai[2] and Ali Ferjani[1,*] 

[1]Department of Biology, Tokyo Gakugei University, Tokyo, Japan; [2]RIKEN Center for Sustainable Resource Science, Yokohama, Japan; [3]Department of Life Sciences, Graduate School of Arts and Sciences, The University of Tokyo, Tokyo, Japan; [4]Graduate School of Bioresources, Mie University, Tsu, Japan

## Original Research Article

**Keywords:**
*Arabidopsis thaliana*; hypocotyl length; peroxisome; IBR10; metabolomics; threonine.

The author responsible for the distribution of materials integral to the findings presented in this article by the policy described in the Instructions for Authors is Ali Ferjani (ferjani@u-gakugei.ac.jp).

**Author for correspondence:**
A. Ferjani,
E-mail: ferjani@u-gakugei.ac.jp

### Abstract

Mobilisation of seed storage reserves is important for seedling establishment in *Arabidopsis*. In this process, sucrose is synthesised from triacylglycerol via core metabolic processes. Mutants with defects in triacylglycerol-to-sucrose conversion display short etiolated seedlings. We found that whereas sucrose content in the *indole-3-butyric acid response 10* (*ibr10*) mutant was significantly reduced, hypocotyl elongation in the dark was unaffected, questioning the role of IBR10 in this process. To dissect the metabolic complexity behind cell elongation, a quantitative-based phenotypic analysis combined with a multi-platform metabolomics approach was applied. We revealed that triacylglycerol and diacylglycerol breakdown were disrupted in *ibr10*, resulting in low sugar content and poor photosynthetic ability. Importantly, batch-learning self-organised map clustering revealed that threonine level was correlated with hypocotyl length. Consistently, exogenous threonine supply stimulated hypocotyl elongation, indicating that sucrose levels are not always correlated with etiolated seedling length, suggesting the contribution of amino acids in this process.

## 1. Introduction

Seedling establishment is an important event in the plant life cycle, orchestrated by transcription factors and enzymes and sustained by a number of low molecular weight compounds. The regulation of hypocotyl development during germination is a complex process, in which many factors must be coordinated temporally and spatially. In *Arabidopsis thaliana* (Arabidopsis, hereafter), sucrose (Suc) is synthesised from the triacylglycerol (TAG) of the oil bodies via a sequence of metabolic reactions including core $\beta$-oxidation, glyoxylate and TCA cycles, and cytosolic gluconeogenesis, and Suc is used to replenish energy supply during seedling development (Graham, 2008).

In a process called lipolysis, SUGAR DEPENDENT1 splits TAG into glycerol and free fatty acids (FAs), which are transported into peroxisomes by the PEROXISOMAL ABC TRANSPORTER 1, where they bind CoA and undergo degradation in the $\beta$-oxidation cycle (Eastmond, 2006; Fulda et al., 2004; Hu et al., 2012). The latter represents a relatively early step in the TAG-to-Suc conversion. A number of loss-of-function mutants defective in this process, such as *ped1*, are unviable unless supplied with an exogenous carbon source, indicating that seedling establishment in Arabidopsis depends on the above metabolic process before photosynthetic capacity is acquired (Germain et al., 2001; Graham, 2008; Hayashi et al., 1998).

Mutations in enzymes catalysing TAG-to-Suc conversion often result in etiolated seedlings with short hypocotyls. More specifically, ISOCITRATE LYASE (ICL; Eastmond et al., 2000) and MALATE SYNTHASE (MLS; Cornah et al., 2004) play important roles in the glyoxylate cycle. In addition, PHOSPHOENOLPYRUVATE CARBOXYKINASE1 (PCK1; Penfield et al., 2004) and, more recently, the vacuolar H$^+$-PPase/FUGU5/AVP1 (Ferjani et al., 2011) were reported to be involved in gluconeogenesis. Collectively, genetic contexts with a molecular lesion in one of the above genes ultimately produce less Suc from TAG during seed germination

(Cornah et al., 2004; Eastmond et al., 2000; Ferjani et al., 2011; Penfield et al., 2004; Takahashi et al., 2017). For instance, whereas *icl-2*, *mls-2*, *pck1-2* and *fugu5-1* mutants exhibit dissimilar metabolic perturbations, albeit with variable penetrance (Ferjani et al., 2018), they share a phenotypic signature, that is, a short hypocotyl in the dark (Ferjani et al., 2011; Takahashi et al., 2017).

A metabolic disorder in TAG-to-Suc conversion concomitant with lowered Suc levels triggers Class II compensation in cotyledons (Takahashi et al., 2017). Compensation is a phenotype whereby decreased cell numbers in determinate organs, such as cotyledons and leaves, trigger a cell size increase (Ferjani et al., 2007; Horiguchi et al., 2006; Horiguchi & Tsukaya, 2011; Tsukaya, 2002; 2008). Therefore, low Suc production often results in reduced cell proliferation activity and thus a reduced cell number in cotyledons due to a low energy state. Importantly, we discovered that *indole-3-butyric acid* (IBA) *response 10* mutants (*ibr10-1*; Zolman et al., 2008; *ibr10-2*; Tabeta et al., 2021) also exhibit Class II compensation. Together with IBR1 and IBR3, IBR10 is involved in the conversion of IBA to indole-3-acetic acid (IAA), and the tuning of auxin homeostasis (Frick & Strader, 2018; Korasick et al., 2013; Spiess et al., 2014; Strader et al., 2011; Zolman et al., 2007; 2008).

Previous studies suggested that the above IBR enzymes catalyse conversion of IBA-CoA to IAA-CoA in a metabolic process resembling peroxisomal $\beta$-oxidation and proposed that they are not dedicated to the production of Suc from TAG (Strader et al., 2011). While *ibr10* exhibited Class II compensation, *ibr1-2* and *ibr3-1* mutants did not show cell number reduction or compensated cell enlargement in cotyledons (Tabeta et al., 2021; Takahashi et al., 2017). Altogether, the above findings suggest a previously unrecognised role of IBR10 in TAG mobilisation, functionally distinct from IBR1 and IBR3 during seedling establishment.

To date, the short hypocotyl phenotype has been ascribed to lowered Suc production during seed germination (Graham, 2008; Strader et al., 2011). While mutants with defects in TAG-to-Suc conversion display short hypocotyls due to limited Suc availability (Cornah et al., 2004; Eastmond et al., 2000; Ferjani et al., 2011; Penfield et al., 2004), hypocotyl elongation of *ibr10* etiolated seedlings was unaffected despite a 40% reduction in Suc content compared with the wild type (WT) (Tabeta et al., 2021). This counter-intuitive phenotype suggests that the length of hypocotyls does not necessarily reflect endogenous Suc content, suggesting the presence of other compounds involved in the regulation of this complex trait.

Discrepancy among *ibr10* and other related mutant phenotypes has led to reconsideration of the physiological function of Suc in seedling establishment (Henninger et al., 2021; Silva et al., 2017). This study, in which a multi-platform metabolome analysis combined with bioinformatics approaches was adopted to unveil the function of IBR10, attempts to tackle this long-standing debate.

## 2. Results

### 2.1. ibr10 mutants exhibit significant metabolic changes compared with ibr1 and ibr3

To address the above question, we performed a wide-target metabolome analysis using gas chromatography–tandem mass spectrometry (GC-QqQ-MS) to assess whether metabolic profiles differ among the *ibr* mutants. To visualise the extent of metabolic changes, we performed principal component analysis (PCA). Our results revealed that while *ibr10-1* was significantly separated compared with the two other *ibr* mutant counterparts in PC1

(46.6%) (Figure 1a), the WT was slightly separated in PC2 (10.1%) and other PC axis combinations (Figure 1a). Interestingly, *ibr1-2* and *ibr3-1* displayed the same behaviour in all PC axis combinations, indicating that these two mutants share common metabolic profiles (Figure 1a). To explore the metabolome data and typical metabolite deviation, we performed hierarchical clustering analysis (HCA) and categorised four different clusters (C1–C4). Consistently, HCA indicated that *ibr1-2* and *ibr3-1* had no obvious metabolic changes (Figure 1b). Interestingly, the largest cluster, C4, was enriched with sugars and organic acids, the amounts of which were significantly inferior in *ibr10-1* compared with the WT (Figure 1b; Supplemental Table S1). Note that a similar trend was observed for some amino acids detected by GC-QqQ-MS (Supplemental Table S1).

In addition, the Venn diagram summarising the metabolic changes among the above *ibr* mutants indicated that only *ibr10* exhibited significant differences in terms of the number of metabolites compared with the WT (Figure 1c). Provided that each mutant had a slightly downregulated rate of IBA-to-IAA conversion in the peroxisome during the early developmental phase (Frick & Strader, 2018; Korasick et al., 2013; Spiess et al., 2014; Strader et al., 2011; Zolman et al., 2007; 2008), changes in 18 selected metabolites in all *ibr* mutants implicate roles in $\beta$-oxidation and/or IBA conversion (Figure 1c). Together, our findings indicated that only *ibr10-1* has a remarkable metabolic alteration among all *ibr* mutants, suggesting a distinct role of IBR10 in seed nutrient reserve mobilisation and a link between such metabolic profiles and the abovementioned cotyledon phenotypes.

### 2.2. Sugar levels were reduced in ibr10 mutants

To investigate the metabolic alterations in *ibr10* mutants, we conducted GC-QqQ-MS wide-target analysis using two different alleles: *ibr10-1* (Zolman et al., 2008) and *ibr10-2* (SALK_201893C; Tabeta et al., 2021). Our data indicated that *ibr10-1* and *ibr10-2* shared ~70% of the total metabolic changes (Supplemental Figure S1). More specifically, 146 metabolites stably detected in the GC-QqQ-MS wide-target analysis were categorised into seven clusters (C1–C7) based on their behaviour (Figure 2a). Among them, the amounts of the C3 metabolites, mostly sugars and organic acids, were lower in the *ibr10* mutants than in the WT (Figure 2b), consistent with the independent data sets described above (see Figure 1b).

Next, we overlaid the profiles of primary metabolites on the corresponding metabolic pathways, focusing on the carbon flow, assuming that this stage of seedling development is characterised by TAG-to-Suc conversion. As shown in Figure 2b, while the levels of metabolites related to the TCA and glyoxylate cycles were either unchanged or slightly reduced, Suc contents were reduced by almost 40% (Figure 2b; Supplemental Table S1). Furthermore, other sugars such as glucose, galactose and fructose were similarly reduced in both *ibr10* alleles (Figure 2b).

Then, the metabolites that were significantly increased or decreased in *ibr10* mutants compared with the WT were further selected by volcano plot analysis. This revealed that adipic acid-2TMS was the most highly accumulated in *ibr10-1* compared with the WT (Figure 2c). Since adipic acid, an important commercial dicarboxylic acid, is a platform chemical that may be linked to $\beta$-oxidation (Skoog et al., 2018), its accumulation in only *ibr10* (Figure 1b) may reflect a TAG-to-Suc mobilisation disorder. In addition, volcano plot analysis revealed that the metabolites that were decreased were mostly sugars and organic acids

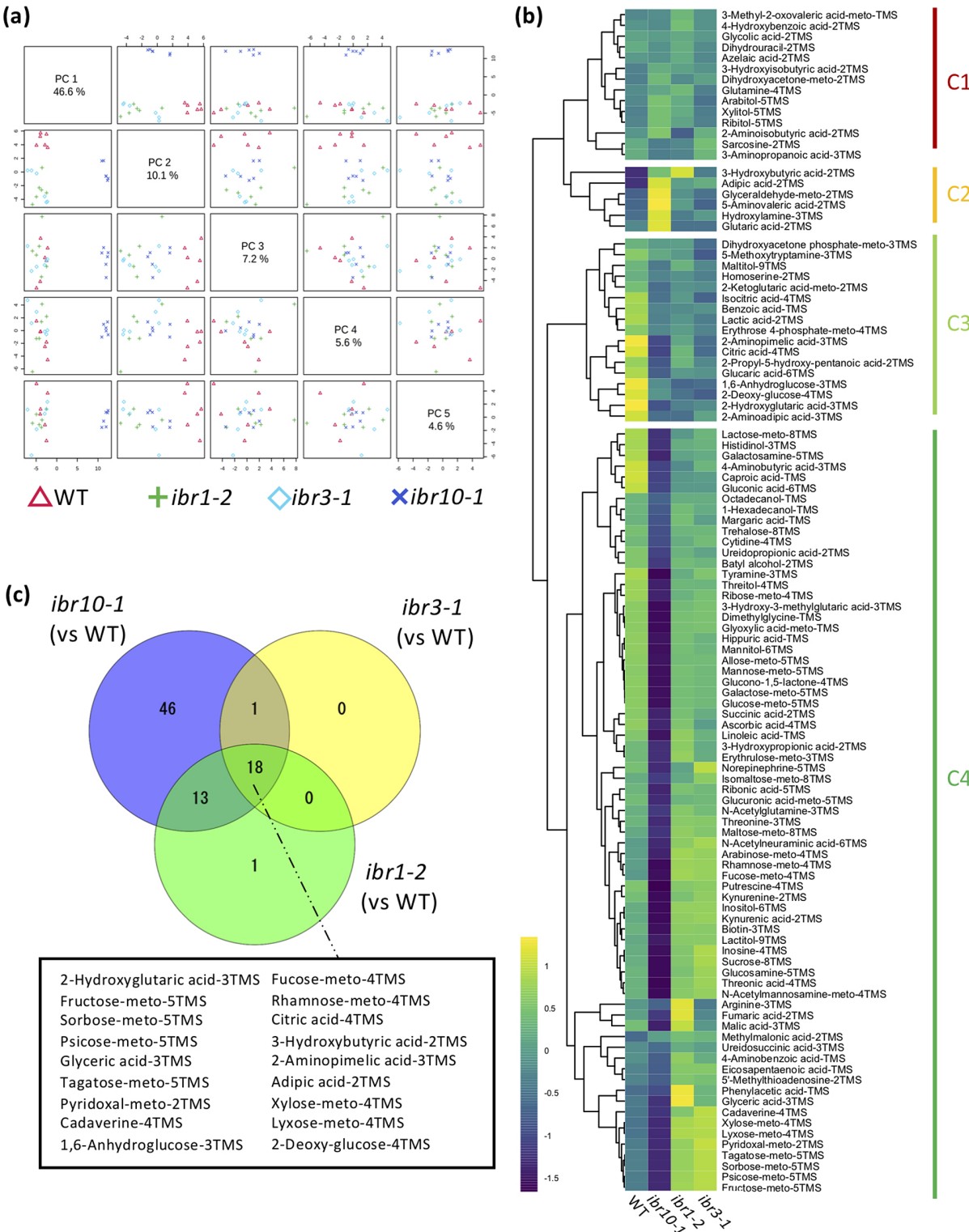

**Fig. 1.** Metabolic changes in TAG-to-Suc conversion in *ibr* mutants. (a) Principal component analysis plot for *ibr* mutants. Metabolome analysis was performed by GC-QqQ-MS using etiolated seedlings at 3 DAI. All 108 metabolites detected by GC-QqQ-MS were used in the PCA (*n* = 6). PC1–PC4 are described. (b) Hierarchical clustering analysis of each genotype. Data are Z-scores of the average content relative to that of the quality control samples (*n* = 6) at 3 DAI. (c) Venn diagrams for each mutant line. Each number represents the metabolites with a significantly changed content compared with the WT, as determined by Student's *t*-test (*n* = 6, *p* < 0.05). Factors shown in the black box are significantly different in all mutants compared with the WT. DAI, days after induction of seed germination.

(Supplemental Table S2). Taken together, these results indicate that *β*-oxidation and carbon flux were specifically affected in *ibr10*, but not in *ibr1-2* or *ibr3-1*. These findings also indicate that the diminished Suc production in *ibr10* mutants is simply due to a lower supply of carbon source from TAG degradation, in agreement with our previous report (Tabeta et al., 2021).

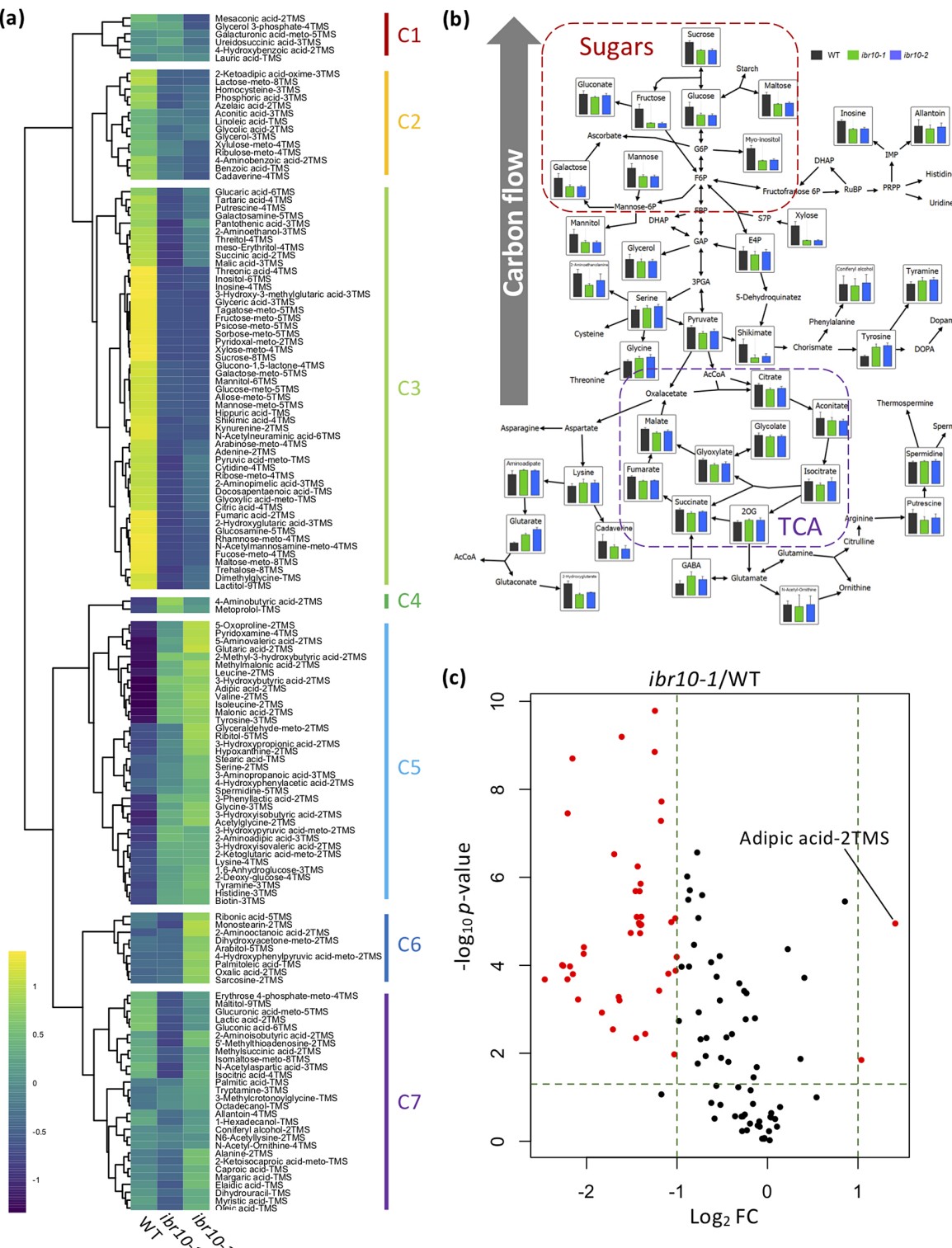

**Fig. 2.** GC-MS/MS metabolome analysis in *ibr10* mutants. (a) Hierarchical clustering analysis of *ibr10-1* and *ibr10-2*. All 146 metabolites detected by GC-QqQ-MS were categorised into seven clusters (C1–C7) based on behaviour. Data are Z-scores of the average content relative to that of the quality control samples (*n* = 6) of 3 DAI etiolated seedlings. (b) Pathway analysis in the *ibr10* mutants. Primary metabolites related to TAG-to-Suc conversion are plotted on the corresponding metabolic pathway. Data are means + standard deviation (SD) (*n* = 6) of 3 DAI etiolated seedlings. (c) Volcano plot for *ibr10-1*. *p*-values determined by Student's *t*-test for WT and *ibr10-1* comparisons. Fold change (FC) values were determined for *ibr10-1* relative to the WT. Red dots represent metabolites significantly changed between the WT and *ibr10-1* (*n* = 6). DAI, days after induction of seed germination.

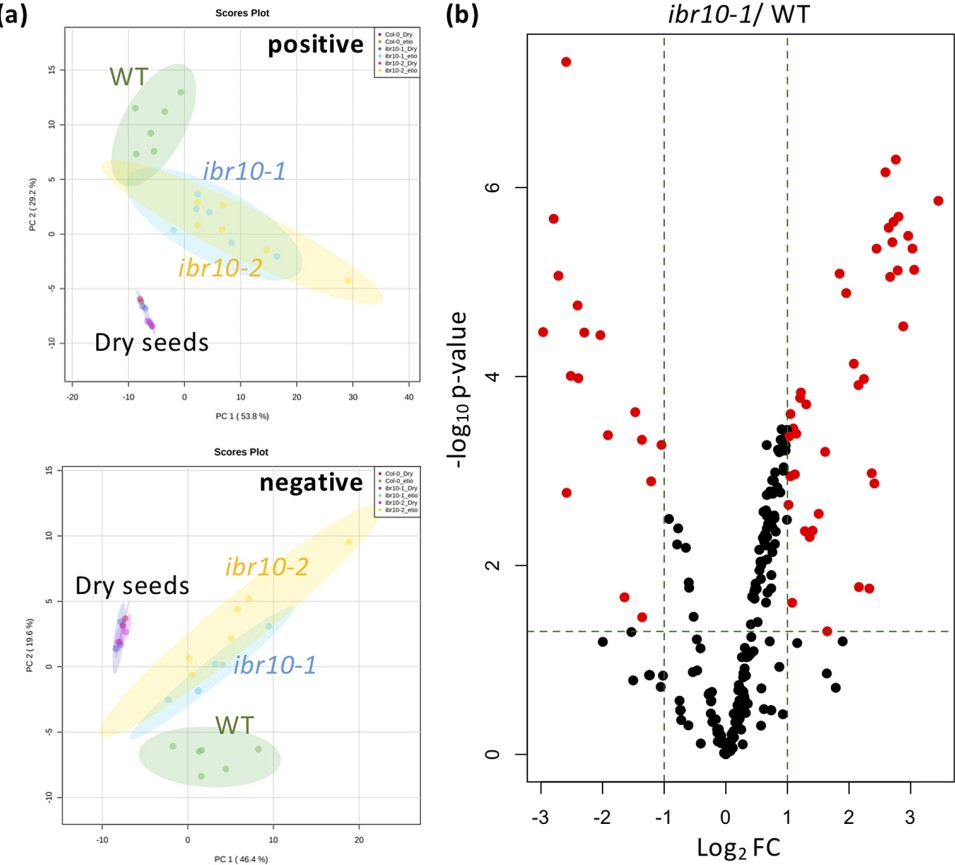

**Fig. 3.** LC-qTOF-MS lipidomic analysis in *ibr10* mutants. (a) PCA plot for the *ibr* mutants. Lipidomic analysis was performed by LC-qTOF-MS non-target analysis using dry seeds (*n* = 3) and etiolated seedlings (*n* = 6) at 3 DAI. Each PCA represents a positive and negative mode of liquid chromatography condition. In total, 156 and 161 metabolites were detected in the positive and negative modes, respectively. (b) Volcano plot for *ibr10-1*. *p*-values were calculated by Student's *t*-test for WT and *ibr10-1* comparisons. Fold change (FC) values were determined for *ibr10-1* relative to the WT. Red dots represent metabolites significantly changed between the WT and *ibr10-1* (*n* = 6). DAI, days after induction of seed germination.

## 2.3. TAG degradation defects were observed in ibr10 mutants

Metabolic profiling indicated that *ibr10* accumulated less sugars and organic acids (Figure 2), suggesting that defects in TAG mobilisation occurred upstream of this pathway. Provided that TAG degradation was nearly complete at 4 DAI (Ferjani et al., 2011), LC-qTOF-MS non-target lipidomic analysis was performed to formally test the above hypothesis at this critical stage.

PCA plots depict the differences between the WT and *ibr10* mutants in both negative and positive modes of LC-qTOF-MS (Figure 3a). Consistently, both *ibr10* mutant alleles showed similar lipidomic changes, confirming the results obtained above by GC-QqQ-MS analysis (Figure 3a). It is of note that the lipid profiles of dry seeds of the WT and two *ibr10* mutant alleles (i.e., prior to seed imbibition) were identical (Figure 3a). Together, these results unambiguously indicated that the metabolic disorders mentioned above occurred after seed imbibition and are likely associated with defects related to TAG degradation and subsequent steps.

Based on the above findings, we next attempted to confirm the changes in lipids in detail in *ibr10*. For this, we performed HCA, which revealed that several lipid species were altered in *ibr10*, particularly in the bottom cluster (in both positive and negative modes), in which TAG, DAG and membrane lipids were enriched (Supplemental Figure S2). The volcano plot further indicated that

several classes of lipids tended to be higher in *ibr10-1* than in the WT (Figure 3b). Next, lipids whose levels were significantly up- or downregulated were selected (Supplemental Table S3). Interestingly, we found that the highly accumulated lipids were almost exclusively TAGs and DAGs, confirming that TAG degradation was severely disrupted. Moreover, levels of monogalactosyldiacylglycerol (MGDG) and digalactosyldiacylglycerol (DGDG), which are targeted mainly to the chloroplast, were markedly low, and phosphatidylethanolamine (PE) and phosphatidylcholine (PC) contents tended to increase in *ibr10* (Supplemental Figure S2). Collectively, these results indicate that TAG mobilisation defects in *ibr10* broadly and significantly affected membrane lipid composition.

It has been reported that a lack of MGDG-to-DGDG conversion disrupts membrane structure in etioplasts (Fujii et al., 2018). Because saturated DGDG is targeted mostly to protoplasts, and their contents were reduced by almost 80% in *ibr10* (Figure 4a), we hypothesised that the transition from etioplast to chloroplast would also be impaired in *ibr10*. Quantification of soil and plant analysis development (SPAD) values, which are correlated with chlorophyll concentration (Ling et al., 2011), suggests that unlike *ibr1-2* and *ibr3-1*, *ibr10* mutants display reduced chlorophyll content during the greening process (Figure 4b). Provided that SPAD values reflect chlorophyll levels via nitrate abundance, this may also

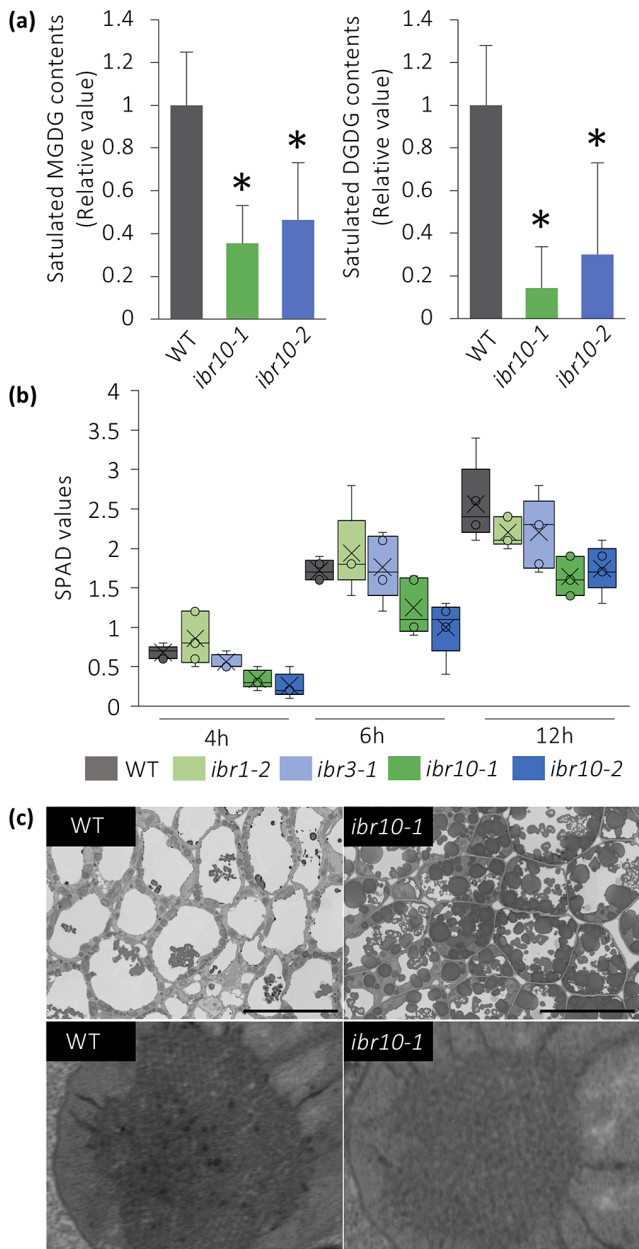

**Fig. 4.** Analysis of etioplast lipid composition and structure. (a) MGDG and DGDG contents of *ibr10* etiolated seedlings. Saturated lipid species were selected from lipidomic data and normalised to WT content. Data are means + SD (*n* = 6; 3 DAI etiolated seedlings). Asterisks indicate mutants with significant differences compared with the WT (Dunnett's test at *p* < 0.05; R ver. 3.5.1). (b) Time-course changes in the SPAD values of *ibr10*. SPAD values were measured at each time point after light treatment (*n* = 5; biological replicates). Lower and upper boxes represent the first and third quartiles, and the line in the boxes represents the median. Each single biological replicate contains 10 technical replicates of 3 DAI etiolated seedlings. (c) FE-SEM images of *ibr10* etiolated seedlings. Upper images represent cell structure, and the corresponding bottom images represent the prolamellar body in etioplasts of 3 DAI etiolate seedlings. Black bar = 20 μm and white bar = 0.5 μm. DAI, days after induction of seed germination; SPAD, soil and plant analysis development.

suggest that TAG degradation defects caused etioplastic transition delay, which in turn led to defects in light-acquired development. Surprisingly, SPAD values measured on rosette leaves were unaffected (Supplemental Figure S3), indicating that chlorophyll content was lower only during the initial developmental phase following light exposure. Although field-emission scanning

electron microscope (FE-SEM) observations support TAG accumulation in *ibr10*, as a significant number of oil bodies remained, the lattice membrane in prolamellar bodies, a unique structure of the etioplast, did not change drastically (Figure 4c). These results suggest that altered lipid composition has a harmful effect during the etioplast–chloroplast transition.

### 2.4. Threonine plays a key role in hypocotyl elongation

Unlike *ibr1-2* and *ibr3-1*, the above results indicated that IBR10 has a non-negligible role in TAG mobilisation (Figure 1). Suc synthesised from TAG is used to produce ATP, which sustains seedling development during heterotrophic growth periods (Graham, 2008). It is widely accepted that the length of hypocotyls in etiolated seedlings is related to endogenous Suc content. However, etiolated seedlings in *ibr10* display a proper length (Strader et al., 2011), suggesting that other metabolite(s) play a key role in this process. While *icl-2*, *mls-2*, *pck1-2* and *fugu5* exhibit shorter hypocotyls in the dark due to metabolic disorders in the TAG-Suc pathway (Cornah et al., 2004; Eastmond et al., 2000; Ferjani et al., 2011; 2018; Penfield et al., 2004), hypocotyl length in these mutants is not strictly correlated with Suc content (Figure 5a; Takahashi et al., 2017; Tabeta et al., 2021). Note that the hypocotyl elongation defect in the above lines is fully restored upon exogenous supply of Suc (Supplemental Figure S4).

To identify candidate metabolites implicated in etiolated seedling length regulation, we conducted a comparative metabolome analysis using WT, *icl-2*, *mls-2*, *pck1-2*, *fugu5-1* and *ibr10-1* as a representative allele. In total, this analysis detected 163 metabolites using the MRM mode of GC-QqQ-MS. Collectively, PCA scatter plots (Figure 5b) and HCA analyses (Supplemental Figure S5) showed remarkable dissimilarities among the above genotypes. Pathway analysis revealed that glucose and fructose contents in *mls-2* were comparable with those in the WT, indicating that these sugars are unlikely to be related to hypocotyl length (Supplemental Figure S5). However, the above approaches alone were insufficient to determine the molecular process underlying hypocotyl length regulation and/or to identify the key metabolites concerned because multiple metabolic processes, namely gluconeogenesis and the glyoxylate and TCA cycles, are affected in the above genotypes (Supplemental Figure S6). We next conducted an additional bioinformatics-based trans-omics analysis to identify such key metabolites.

To determine the relationships between phenotypes and metabolome data, batch-learning self-organised map (BL-SOM) analysis (Abe et al., 2003; Kanaya et al., 2001), an approach that was first applied during metabolome and transcriptome overlay analysis (Hirai et al., 2004; 2005), was employed. Here, we used metabolome data and quantitative hypocotyl length data. Each mutant line displayed a unique fingerprint (Figure 5c); we found that length was categorised into a cluster (6,6) that included three metabolites: cytidine, ribose and threonine [(Thr); Figure 5c,d]. Interestingly, Suc was categorised into cluster (7,6), indicating that the Suc content was not correlated with hypocotyl length under our conditions.

Among the three metabolites, provided that ribose and cytidine were scarcely detected in this extraction method, the effect of the exogenous supply of Thr was assessed first at different concentrations (100 nM–1 mM). Importantly, L-Thr consistently and significantly promoted the elongation of etiolated seedling hypocotyls in the WT, at all concentrations tested so far (Supplemental Figure S7a). Surprisingly, the exogenous supply of Gly, Ser,

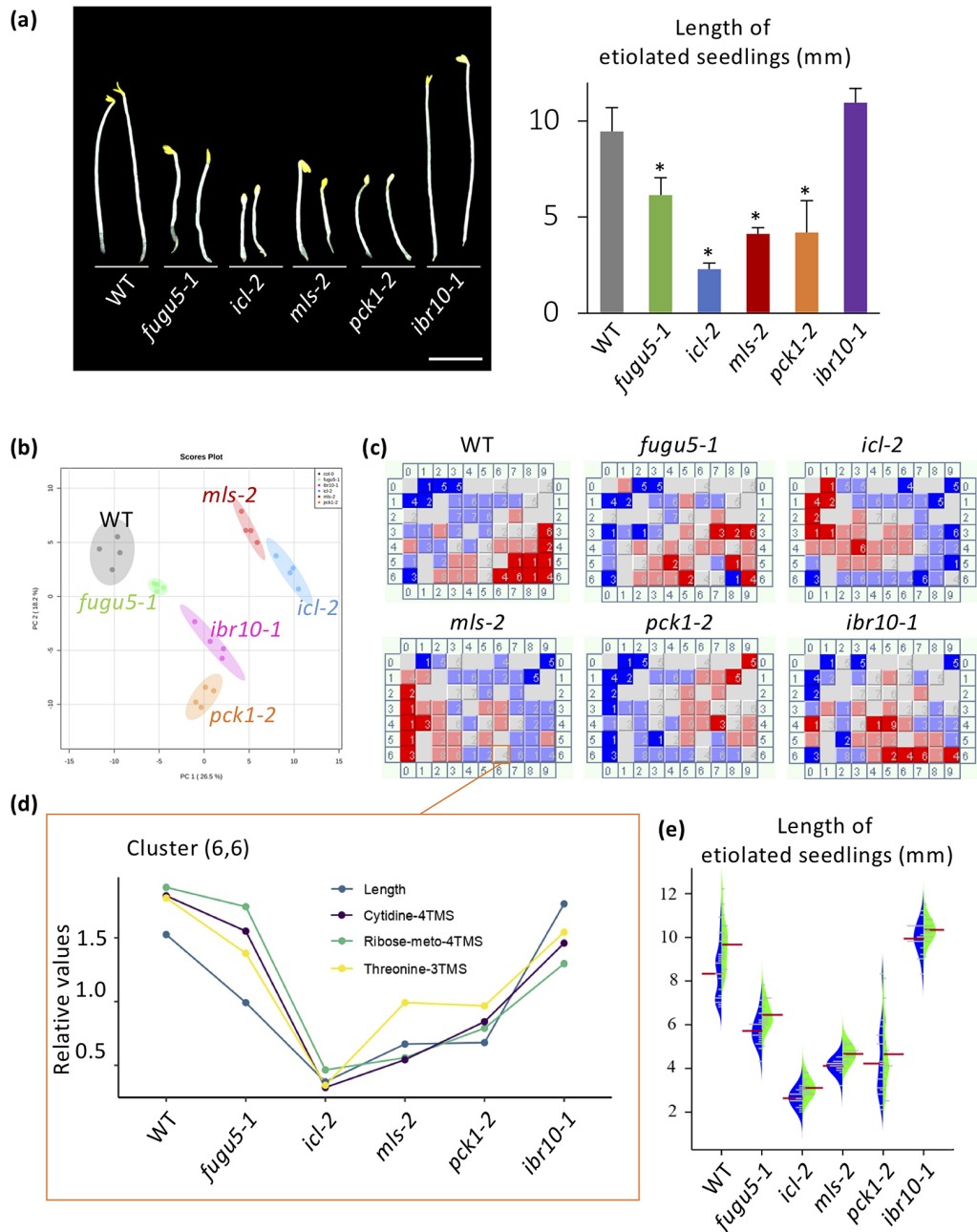

**Fig. 5.** Key metabolites with contents correlated with hypocotyl length. (a) Phenotypes of etiolated seedlings grown on Suc-free MS medium. Photographs were taken at 3 DAI (left panel, bar = 3 mm). The bar plot presents the means + SD of 3 DAI etiolated seedlings (*n* = 20). Asterisks indicate mutants with significant differences compared with the WT (Dunnett's test at *p* < 0.05; R ver. 3.5.1). (b) PCA plot of Class II compensation mutants. Overall, 163 metabolites were detected by GC-QqQ-MS and analysed. Each plot represents samples (*n* = 4) from 3 DAI etiolated seedlings. (c) BL-SOM analysis of etiolated seedlings. All metabolites were categorised into 70 clusters based the elongation pattern of each mutant. Each box contains metabolites with similar patterns. Blue and red represent low and high amounts compared with the average values, respectively. (d) Bar graph of factors categorised in the cluster (6,6). Data indicate relative contents in each mutant line. (e) Beanplot of etiolated seedling length. Red bars represent means. Blue blocks represent MS medium data, and green blocks represent MS + 10 μM Thr data of 3 DAI etiolated seedlings (*n* = 20). Etiolated seedlings were significantly longer on MS + 10 μM Thr compared to Thr-free MS medium in all indicated genotypes, except the *pck1-2* mutant (Student's *t*-test at *p* < 0.05). DAI, days after induction of seed germination.

Cys, Lys, Glu, Asp and Phe at 100 μM had no promotive effect on the elongation of etiolated seedling hypocotyls (Supplemental Figure S7b). Finally, when exogenously supplied, L-Thr enhanced hypocotyl elongation in the WT and all mutant lines (Figure 5e).

## 3. Discussion

Despite the abundance of data, our understanding of the mechanism of etiolated seedling elongation remains elusive. From

seed imbibition until the acquisition of photosynthetic ability, TAG-derived Suc is a vital carbon and energy source that fuels seedling elongation in Arabidopsis (Graham, 2008). We previously reported that *ibr10* mutants exhibit normal etiolated seedlings, despite having a severely decreased Suc content (Tabeta et al., 2021). This discrepancy contradicts the widespread belief that Suc content is directly linked to etiolated seedling length.

While the roles of IBR1, IBR3 and IBR10 in IAA homoeostasis have been well documented (Frick & Strader, 2018; Korasick et al.,

2013; Spiess et al., 2014; Strader et al., 2011), their function in TAG mobilisation has been overlooked. Surprisingly, among the three *ibr* mutants, only the *ibr10* mutant exhibited Class II compensation (Tabeta et al., 2021). Because Class II compensation is exclusively associated with the partial failure of Suc production de novo from TAG, we extensively investigated this pathway in the *ibr* background.

### 3.1. IBR10 has an important role in TAG-to-Suc conversion

Arabidopsis mutants, even with a partial defect in Suc production from TAG, typically display short hypocotyls in the dark (Cornah et al., 2004; Eastmond et al., 2000; Ferjani et al., 2011; 2018; Penfield et al., 2004). However, *ibr10* mutants in which Suc content was reduced by almost 40% did not follow this rule (Tabeta et al., 2021), questioning the validity of the correlation between hypocotyl length and Suc. To this end, metabolome analysis was performed to identify novel factors promoting hypocotyl elongation that may have been overlooked using conventional methods.

LC-qTOF-MS and GC-QqQ-MS analyses of major metabolites that occur during TAG-to-Suc conversion and found that IBR10 plays a critical role. Our data revealed that the levels of several metabolites in the TCA and glyoxylate cycles, as well as a number of organic acids related to these pathways, were reduced in *ibr10* mutants (Figure 2b). Moreover, the overall sugar content was also reduced (Figure 2b), indicating that the loss of IBR10 activity affected TAG mobilisation. Consistently, lipidomics revealed that TAGs and DAGs remained at high levels (Figure 3; Supplemental Figure S2), suggesting that the process occurring from TAG degradation to the TCA and glyoxylate cycles is compromised in *ibr10*.

Whereas previous studies reported that IBR10 is localised in plant peroxisomes together with other enzymes involved in β-oxidation (Reumann et al., 2007), its involvement in TAG mobilisation has been obscured by the normal hypocotyl length of its etiolated seedlings (Strader et al., 2010; 2011). Against the above background, we propose that IBR10 is indeed involved in β-oxidation, directly or indirectly, given that its loss-of-function led to TAG and DAG degradation defects. More specifically, FAs released from the TAG of the oil bodies via lipase activity are transported into peroxisomes, where they are transformed to acyl-CoA to enter the core β-oxidation cycle (Eastmond, 2006; Fulda et al., 2004; Graham, 2008; Hu et al., 2012). As adipic acid is a FA related to β-oxidation, the high accumulation of TAGs and DAGs in *ibr10* (Figure 2) may have affected closely related reactions and/or transport of FAs, again suggesting a major contribution of IBR10 to lipid mobilisation. However, we were unable to identify the substrate of IBR10 in our omics-based analysis; this should be addressed in future work.

ENOYL-CoA HYDRATASE2, which is involved in both β-oxidation and TAG-Suc mobilisation (Katano et al., 2016; Li et al., 2019; Strader et al., 2011), may share similar functions with IBR10. It was interesting to find that *ibr1-2* and *ibr3-1*, reported to be dedicated to IBA conversion, also displayed slight metabolic fluctuations compared with the WT (Figure 1), pointing to a yet unidentified connection between the two pathways. The peroxisome is an essential organelle in plant cells in which several vital chemical reactions take place. This study adds insight to our current understanding of IBRs and the peroxisomal metabolic network.

### 3.2. Missing phenotypes caused by lack of IBR10 enzyme were identified by homeostatic metabolic networks

Lipid mobilisation is an essential metabolic event during the early phase of plant developmental, not only as a source of energy but also to acquire photosynthetic ability, which involves chloroplast ultrastructural changes during the greening process. Mutants with defects in MGDG-to-DGDG conversion have a defect in etioplasts, which leads to lowered photosynthetic ability, indicating the importance of DGDG content in the transition into protoplasts and light-acquired development (Fujii et al., 2018). In *ibr10*, sugars levels were reduced by ~40% due to peroxisomal metabolic disorder, which may also reduce the UDP-galactose used during DAG-to-MGDG conversion. Recently, Kozuka and colleagues reported that *icl-2* and *mls-2* mutants display low sugar contents and have a photosynthesis defect at an early developmental phase (Kozuka et al., 2020), suggesting TAG breakdown defects affect galactolipid synthesis.

In the *ibr10* mutants, a defect in lipid mobilisation likely reduced MGDG and DGDG contents by ~40% and 20–30%, respectively, mimicking the lipid composition reported previously in Fujii et al. (2018). Unexpectedly, although *ibr10* exhibited an altered membrane lipid composition and low photosynthesis ability, etioplast ultrastructure was not disrupted (Figures 3b and 4b; Supplemental Figure S2). Our metabolome analysis and previous research showed that the metabolic alteration starts after 2–3 DAI (Figure 3a; Tabeta et al., 2021), indicating that TAG degradation defects after seed imbibition and their impact on MGDG and DGDG biosynthesis are exclusively post-germinative. Altogether, TAG jams at the seed imbibition stage slowed lipid mobilisation and affected galactolipids, resulting in a transition delay to the autotrophic regime, likely due to low chlorophyll contents.

Lipid homoeostasis of MGDG, DGDG, PE and PC has been discussed previously (Essigmann et al., 1998; Hartel & Benning, 2000; Tjellstrom et al., 2008). Our data are consistent with the above trend under the assumption that PE and PC levels were increased in *ibr10* to compensate for the reduction in galactolipids (Supplemental Figure S2). Nevertheless, although oil bodies were not fully converted into sugars in *ibr10*, this metabolic disruption did not severely affect hypocotyl elongation. This may indicate that central metabolic flow was somehow adjusted. Arabidopsis seed storage proteins represent a second gateway fueling seedling establishment (Eastmond et al., 2015). It is also possible that *ibr10* mutant can generate longer hypocotyls than the other mutants because it diverts its energy from chlorophyll synthesis and chloroplast biogenesis to hypocotyl elongation. Therefore, these alternative pathways may have partially helped replenish energy shortage during *ibr10* seed germination, as *ibr10* failed to fully use TAG stores. Collectively, homoeostatic coordination among the different lipid species and central metabolism may have led to a new steady state compensating for the deficit in the *ibr10* background, preventing a visible phenotype.

### 3.3. Threonine as a potential regulator of hypocotyl elongation

As mentioned above, our phenotypic analysis revealed that Suc levels do not always allow prediction of hypocotyl length in etiolated seedlings. Indeed, Suc was not categorised within the same cluster as etiolated seedling length (Figure 5c), and *ibr10* displayed normal hypocotyl length (Tabeta et al., 2021). Taken together, these results suggest that other factors may control seedling development.

BL-SOM analysis identified candidate metabolites with a straightforward relationship with hypocotyl length (Figure 5c,d). Among the metabolites identified, the contents of the amino acid threonine (Thr) were positively correlated with hypocotyl length, and exogenous supply of Thr at a relatively low concentration (10 μM) significantly boosted hypocotyl elongation (Figure 5e). It is worth noting that our targeted GC-QqQ-MS analysis detected only 163 metabolites, which is a relatively low number when considering the rich plant phytochemical diversity. For instance, Arabidopsis alone has been reported to possess nearly 3,030 different metabolites, a number that may double in the future (Hawkins et al., 2021). This may suggest that other metabolites or Thr-related compounds not detected here may have a relatively higher biological activity in stimulating hypocotyl elongation. In other words, our findings suggest that not only sugars but also various metabolites, such as amino acids and/or organic acids, must be tightly regulated during organogenesis (Kawade et al., 2020; Nakayama et al., 2022).

### 3.4. The importance of quantitative approaches in plant development

IBR1 and IBR3 have potential functions with IBR10 in IBA-IAA conversion (Strader et al., 2010; 2011; Zolman et al., 2007; 2008); etiolated seedlings in the above mutants were not affected (Zolman et al., 2008). We recently reported that *ibr10* exhibits Class II compensation, which is observed in mutants with partially defective TAG-to-Suc conversion, indicating that IBR10 has a previously unrecognised function (Tabeta et al., 2021; Takahashi et al., 2017). This is likely because *ibr10* has no visible phenotype, as it is indistinguishable from the WT.

Although the holistic contributions of carbohydrates to etiolated seedling establishment and hypocotyl elongation have been investigated extensively, the molecular processes behind this important trait have not been fully determined. For decades, most studies suggested a direct link between Suc concentration and hypocotyl length (Graham, 2008; Strader et al., 2011). First, against the above background, this study provides a conceptual overview of how such questions should be approached. Second, our previous studies (Tabeta et al., 2021; Takahashi et al., 2017), together with the quantitative trans-omic approach adopted in this study, unambiguously show that Thr promotes hypocotyl elongation in the dark. This is in full agreement with a previous report that Arabidopsis uses two gluconeogenic gateways for organic acids to fuel seedling establishment (Eastmond et al., 2015).

More specifically, it has been thought for a long time that the gluconeogenic route in eukaryotes uses organic acid intermediates to produce sugars in a PCK-dependent manner. Surprisingly, radiolabelling showed that while PCK allows de novo synthesis of sugars from dicarboxylic acids, products of lipid breakdown, the cytosolic enzyme orthophosphate dikinase allows sugars to be produced from pyruvate, a major product of protein breakdown (Eastmond et al., 2015). As Ala, Cys, Gly, Trp and Thr from protein reserves represent a major source of pyruvate (Eastmond et al., 2015), our findings imply that hypocotyl elongation promoted by exogenous Thr supply reflects the role of storage proteins in nurturing seedlings during a skotomorphogenic developmental programme. In other words, while our data indicate that Thr can act both as an alternative energy source and a carbon source during hypocotyl elongation, further experiments are needed to determine the full molecular mechanism. This study reveals a new role of Thr in cell elongation, allowing for the first time the uncoupling of certain etiolated mechanisms using a quantitative metabolomic approach. Finally, our findings demonstrate the importance of quantitative plant biology approaches (Autran et al., 2021) in depicting and visualising the unseen side(s) of plant development and provide meaningful insight into uncharacterised regulatory mechanisms.

## 4. Methods

### 4.1. Plant materials and growth conditions

The WT plant used in this study was Columbia-0 (Col-0), and all the mutants were based on the Col-0 background. *ibr1–2*, *ibr3–1* and *ibr10–1* seeds were a gift from Professor Bonnie Bartel (Rice University). *icl–2*, *mls–2* and *pck1–2* mutant seeds were a gift from Professor Ian Graham (The University of York). *ibr10-2* was obtained from Arabidopsis Biological Resource Center (ABRC/The Ohio State University).

Sterilised seeds were sown on Suc-free Murashige and Skoog (MS) medium (Wako Pure Chemical) or MS medium with 2% (w/v) Suc where indicated. 0.1% (w/v) 2-(N-morpholino) ethane-sulfonic acid (MES) was added, the pH was adjusted to 5.8 with KOH, and then the medium was solidified with 0.2–0.5% (w/v) gellan gum (Murashige & Skoog, 1962) in order to determine the effects of medium composition on phenotype. Stock solutions of L-threonine (Thr) and L-serine (Ser; Sigma-Aldrich), or glycine (Gly), L-Cysteine (Cys), L(+)-Lysine (Lys), L(+)-Glutamine (Glu), L-Aspartic Acid (Asp) and L(-)-Phenylalanine (Phe) (Wako Pure Chemical) were filter sterilised and added to Suc-free MS medium at the indicated final concentrations. The seeds were sown on the MS plates, which were then stored at 4°C in the dark for 3 days. After cold treatment, the seedlings were grown in the dark for the designated durations. *icl–2*, *mls–2*, *pck1–2*, *ibr1–2*, *ibr3–1*, *ibr10–1* and *ibr10–2* were genotyped and characterised as described previously (Cornah et al., 2004; Eastmond et al., 2000; Penfield et al., 2004; Strader et al., 2011; Tabeta et al., 2021; Takahashi et al., 2017).

### 4.2. Microscopy

Photographs of gross plant phenotypes at 3 DAI were taken using a stereoscopic microscope (M165FC; Leica Microsystems) connected to a CCD camera (DFC300FX; Leica Microsystems).

For electron microscopy, samples were fixed in 4% paraformaldehyde and 2% glutaraldehyde in 50 mM sodium cacodylate buffer (pH 7.4) for 1 hr in the dark (covered with aluminum foil) and for 1.5 hr in the light at room temperature. Then, the samples were post-fixed with 1% osmium tetroxide in 50 mM cacodylate buffer for 2 hr at room temperature. After dehydration in a graded methanol series (25, 50, 75, 90 and 100%), the samples were embedded in Epon812 resin (TAAB). Ultrathin sections (100 nm) were generated using a diamond knife on an ultramicrotome (Leica EM UC7, Leica Microsystems) and placed on a glass slide. The sections were stained with 0.4% uranyl acetate for 12 min followed by lead stain solution (Sigma-Aldrich) for 3 min and coated with osmium under an osmium coater (HPC-1SW, Vacuum Device). The sections were observed using the FE-SEM SU8220 (Hitachi High Technology) with an yttrium aluminum garnet backscattered electron detector at an accelerating voltage of 5 kV.

### 4.3. Wide-target metabolome analysis of GC-QqQ-MS

Etiolated seedlings at 3 DAI were collected in one tube in liquid nitrogen and freeze-dried. The samples were extracted using a bead

shocker in a 2 mL tube with 5 mm zirconia beads and 80% MeOH for 2 min at 1,000 rpm (Shake Master NEO, Biomedical Sciences). The extracted solutions were centrifuged at $10^4$ g for 1 min, and 100 μL centrifuged solution and 10 μL 0.2 mg/mL Adonitol (Internal Standard; I.S.) were dispensed in a 1.5 mL tube. After drying the solution using a centrifuge evaporator (Speed vac, Thermo), 100 μL Mox reagent (2% methoxyamine in pyridine, Thermo) was added to the 1.5 mL tube, and the metabolites were methoxylated at 30°C and 1,200 rpm for approximately 6 hr using a Thermo shaker (BSR-MSC100, Biomedical Sciences). After methoxylation, 50 μL 1% v/v of trimethylchlorosilane (TMS, Thermo) was added to the 1.5 mL tube. For TMS derivatisation, the mixture was incubated for 30 min at 1,200 rpm at 37°C as described above.

Finally, 50 μL of the derivatised samples was dispensed in vials for GC-QqQ-MS analysis (AOC-5000 Plus with GCMS-TQ8040, Shimadzu Corporation). Raw data collection was performed using the GCMS software solution (Shimadzu Corporation). Calculation of the peak area values was conducted using MRMPROBS (Tsugawa et al., 2013; 2014a; 2014b). Peak areas were normalised using a quality control sample and LOWESS/Spline normalisation tool (Tsugawa et al., 2014a). Detailed GC-MS/MS parameters were as described previously (Tabeta et al., 2021). MRM transition information is shown in Supplemental Table S4.

### 4.4. Lipidome analysis using LC-qTOF-MS

0.8 mg (DW) of mature seeds (50 dry seeds per sample) or 50 etiolated seedlings grown under complete darkness on Suc-free MS medium were used as the starting material for lipidomic analysis. Dry seeds (three biological replicates) or etiolated seedlings (six biological replicates) placed in a 2-mL centrifuge tube were mixed with an 800-fold volume of extraction solvent [methyl *tert*-butyl ether/methanol = 3/1 (v/v) containing 1 μM 1,2-didecanoyl-sn-glycero-3-phosphocholine, Sigma-Aldrich], and milled by shaking at 900 rpm at 4°C for 5 min on the Shake Master Neo (BMS, Tokyo, Japan) using zirconia beads. A 200-fold volume of water was added to the homogenate. After vigorous stirring on a vortex mixer and dark incubation for 15 min on ice, the homogenate was centrifuged at 1,000 × g for 5 min. The upper layer (200 μL) was transferred to a new 1.5-mL microcentrifuge tube. The organic phase was evaporated to dryness using a centrifugal concentrator (ThermoSavant SPD2010, Thermo Fisher Scientific) at room temperature. The residue was dissolved in 250 μL ethanol and centrifuged at 10,000 × g for 15 min. Then, 200 μL of the supernatant was transferred to a vial with a glass insert for performing LC-MS/MS analysis. The method for LC-MS analysis was described previously (Okazaki & Saito, 2018).

### 4.5. SPAD measurement

Plants were cultured as described above, and etiolated seedlings were placed under the light to develop etioplasts into chloroplasts. SPAD values were recorded at 4, 6 and 12 hr using the cotyledons of etiolated seedlings (Konica-Minolta, Japan, SPAD-502 plus). Single data described as one replicate contained 10–11 technical replicates using 10 etiolated seedlings.

### 4.6. Bioinformatics analyses

Statistical analyses were performed using Student's *t*-test or Dennett's test (R ver. 3.5.1; R Core Team, 2018). Multiple comparisons

were performed using the multcomp package (Hothorn et al., 2008). PCA plots were calculated and designed in MetaboAnalyst4 (Chong et al., 2019). Venn diagrams were designed using Venny 2.1 (Oliveros, 2007–2015). HCAs were performed using the pheatmap package (Kolde, 2019). Pathway analysis was performed using VANTED software (Rohn et al., 2012) and the KEGG database (Kanehisa, 2002; Kanehisa & Goto, 2000; Okuda et al., 2008). BL-SOM analysis was performed using the simple SOM programme (Abe et al., 2003; Kanaya et al., 2001). Bean plots were performed using the beanplot package (Kampstra, 2008). Calculation of $log_{10}$ *p*-values and $log_2$ fold change values for the volcano plots, *p*-values for the Venn diagram and autoscaling sample data for HCA were conducted using MetaboAnalyst4 and 5 (Chong et al., 2019).

### Acknowledgements

We thank Kouji Takano (RIKEN Center for Sustainable Resource Science) for his technical assistance with LC-qTOF-MS analysis. We thank Dr. Kensuke Kawade (NIBB) and Dr. Takashi Nobusawa (Hiroshima University) for critical reading of the manuscript and valuable comments. We thank Prof. Ian Graham (The University of York) and Dr. Alison Gilday (The University of York) for providing *icl–2*, *mls–2* and *pck1–2* mutant seeds. We also thank Professor Bonnie Bartel (Rice University) and Dr. Lucia C. Strader (Duke University) for providing *ibr1-2*, *ibr3-1* and *ibr10-1* seeds.

**Financial support.** This work was supported by Grant-in-Aid for Scientific Research (B) (16H04803 to AF); Grant-in-Aid for Scientific Research on Innovative Areas (25113002 to AF; 25113010 to MYH); Grant-in-Aid for Scientific Research on Innovative Areas (18H05487 to AF). H. Tabeta is a recipient of a Research Fellowship for Young Scientists (DC1) (20J20901).

**Conflicts of interest.** The authors declare no conflicts of interest.

**Authorship contributions.** H.T. performed the experiments, collected and analysed the data, and drafted the article. Y.H. and Y.O. performed lipidomics data collection. K.T, M.W. and M.S. performed FE-SEM analysis. K.S. and M.Y.H. directed and funded the study. A.F. conceived the project, designed, supervised, funded the study and wrote the paper with input from all co-authors. All authors read and approved the final manuscript.

**Data availability statement.** The raw data supporting the conclusions of this article will be made available by the authors, without undue reservation, to any qualified researcher.

**Supplementary Materials.** To view supplementary material for this article, please visit http://doi.org/10.1017/qpb.2022.19.

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
