## [Reviewer Report]

July 2, 2022

Editor-in-Chief,

Dear Prof. Olivier Hamant,

Please consider our manuscript entitled “Skotomorphogenesis exploits threonine to promote hypocotyl elongation” for publication, as an Original Research Article, in Quantitative Plant Biology.

Plant Seedlings kept in darkness adopt a skotomorphogenic developmental program, in which allocation of resources is typically directed toward hypocotyl elongation at the expense of cotyledon and root development. Energy production from seed nutrient reserves is crucial for transition of a plant seedling from a heterotrophic to autotrophic regime. In Arabidopsis thaliana, sucrose (Suc) is synthesized from the triacylglycerol (TAG) of the oil bodies via a sequence of metabolic reactions including core β-oxidation, glyoxylate and TCA cycles, and cytosolic gluconeogenesis, and Suc is used to replenish energy supply during seedling development.

To date, the short hypocotyl phenotype has been ascribed to lowered Suc production during seed germination. While mutants with defects in TAG-to-Suc conversion display short hypocotyls due to limited Suc availability, we previously reported that hypocotyl elongation of *indole-3-butyric acid response 10* (*ibr10*) etiolated seedlings was unaffected despite a 40% reduction in Suc content compared with the wild type (Tabeta et al., 2021 *PLOS Genetics*). This counter-intuitive phenotype suggests that the length of hypocotyls does not necessarily reflect endogenous Suc content, hinting to the presence of other compounds involved in the regulation of this complex trait. This study, in which a multi-platform metabolome analysis combined with bioinformatics approaches was adopted to unveil the function of IBR10, attempts to tackle this long-standing debate.

Here, to dissect the metabolic complexity behind cell elongation, a quantitative-based phenotypic analysis combined with a multi-platform metabolomics approach was applied. We revealed that TAG and diacylglycerol breakdown was disrupted in the *ibr10* mutant, resulting in low sugar content and poor photosynthetic ability. Importantly, batch-learning self-organized map clustering revealed that the endogenous threonine level was correlated with hypocotyl length. Consistently, exogenous threonine supply stimulated hypocotyl elongation, indicating that Suc levels are not always correlated with etiolated seedling length, suggesting the contribution of amino acids in this developmental process.

As you may have appreciated, this study reveals a new role of the amino acid threonine in cell elongation, allowing for the first time the uncoupling of certain etiolated mechanisms using a quantitative metabolic approach. Finally, our findings demonstrate the importance of quantitative plant biology approaches in depicting and visualizing the unseen side(s) of plant development and provide meaningful insight into uncharacterized regulatory mechanisms.

This manuscript is the result of a collaboration between several research groups. Dr. Masami Hirai is an expert in plant metabolomics. Dr. Yozo Okazaki is an expert in plant lipidomics, and Dr. Kiminori Toyooka is an expert of electron microscopy.

We believe that our findings will interest a range of scientists who are interested in developmental biology, metabolomics and the related fields of cell biology.

The submitted version has been approved by all authors. This manuscript has not been submitted to another journal, and none of the results have been published elsewhere. The manuscript has been proofread by two native English speakers.

We believe that our manuscript reports findings of key importance about the metabolic complexity behind cell elongation during Skotomorphogenesis from interdisciplinary and quantitative angles. Please do not hesitate to contact us if you require any additional information.

I look forward to hearing from you.

Sincerely,

Ali FERJANI

Ph. D (Associate Professor)

Tokyo Gakugei University

---

## [Reviewer Report]

*Comments to Author*: This manuscript by Tabeta et al. reports that sucrose level s are not always correlated with etiolated seedling development, and that one of amino acids, threonine, contributes hypocotyl elongation after post germination. To derive the above results, the authors performed multi-omics analyses including metablomics and lipidomics, and electron microscopic observation using Arabidopsis ibr10 mutants. Their results looks sound and confirmatory, and the reviewer thinks the information in this manuscript contribues to the understanding of skotomorphogenesis. However, the current manuscript should be reviesed to make the data more solid. Below are some of the comments, which should be considered to improve the manuscript.

The reviewer agrees with the usefulness of the multi-omics approach performed taken by the authors, and believe that it will be one of the important approaches in the future life science research.

Specific comments

(1) The first and second paragraphs in Result section (lines 100-111) should be moved to Introduction or Discussion section.

(2) Discussion section is too much of a duplication of Introduction and Results sections. The description should be more considered.

(3) The authors used Arabidopsis ibr mutants, especially ibr10, as plant materials. However, as written in Discussion section, IBR10 is revealed to play a lesser role in beta-oxidation process. There, the reviewer feels that the relationship with IBR10 and energy production via beta-oxidation in peroxisomes is obscure. Can the authors provide any further comments about this point?

(4) Multi-omics analyses revealed that threonine is also involved in regulation of hypocotyl elongation. In fact, the addition of exogeneous threonine has been shown to extend the hypocotyls (Figure 5E). The authors should add the data using other amino acids to Figure 5E to demonstrate the specificity of threonine.

(5) Regarding the above comment, can the authors should add more information why threonine is involved in hypocotyl elongation? This is the crucial point, which is related to this paper.

(6) The authors should add the information for growth condition in Figure legend for Figure 5A. Does the medium contain sucrose? If this experiment was performed using the medium without sucrose, a picture of the seedlings under normal growth condition using the medium containing sucrose should be added as a control.

(7) The authors discussed the contribution of protein storage proteins for generation of energy in seedling establishment process in Discussion section (lines 309-314). Why not include data from proteome analysis in wild type and ibr10 mutants? In addition, can the author add the data to show the morphology and numbers of protein bodies using FE-SEM observation in Figure 4C?

(8) In Methods section, the description of centrifugation conditions, such g to rpm, is mixed and should be unified.

Minor comments

(9) Abbreviations of journal names in some references (Autran et al., Hirai et al. (2014) etc) are not uniformly written.

(10) The word of ‘beta’ in some references was depicted in symbol.

(11) The initial letters of all words in titles of some references (Katano et al, Kawade et al. etc) are capitalized.

(12) Replace ‘H. Tabata’ to ‘H.T.’ in Author contribution section.

---

## [Reviewer Report]

*Comments to Author*: Tabeta et al. previously reported an interesting feature of an Arabidopsis ibr10 mutant, i.e. unaffected hypocotyl elongation of etiolated seedlings despite defects in mobilization of storage lipids into sucrose (Tabeta et al. [52]. PLOS Genet 17: e1009674). This manuscript describes their follow-up work to account for this phenomenon by using a quantitative metabolomics approach. It was shown that ibr10 mutants exhibited lower levels of sugars and organic acids but higher levels of diacylglycerols, triacylglycerols and membrane lipids. Also, the accumulation of adipic acid is indicative of impaired beta-oxidation. Their work has two breakthrough discoveries: (1) Hypocotyl length of etiolated seedlings is not correlated with sucrose content, challenging such a belief held for decades; (2) Instead, the hypocotyl length is correlated with content of the amino acid threonine (Thr), which boosted hypocotyl elongation upon exogenous application. The authors believe that this work exemplifies the importance of quantitative approaches in discovering unseen sides of plant development/uncharacterized regulatory mechanisms. This approach appears to be appropriate as the first attempt to address their research questions. However, it would be better if the conclusions are supported by other data, not solely based on the metabolic profiles.

Specific comments:

1) The accumulation of triacylglycerols in ibr10 seedlings seems to support the finding that beta-oxidation is impaired to supply sucrose. However, non-storage lipids (e.g. diacylglycerol, membrane lipids) are also upregulated. Whether the accumulation of lipids truly reflects a defect in mobilization of storage lipids depends on an important control, i.e. lipid content of mature seeds. Although there is one statement stating that the lipid profiles of dry seeds of WT vs ibr10 were identical (Line 173-174), did the authors observe from their electron micrographs (Fig. 4C) that there is no difference in oil body content of mature WT vs mutant seeds while oil bodies gradually disappear during the course of seed imbibition in WT but not in mutant? Similarly, was the change in MGDG and DGDG correlated with the amount of plastids? The method of lipidomic analysis is provided for mature seeds (section 4.4). Was the same method used for lipid extraction from seedlings? It is important to confirm that the altered lipid profiles of mutant seedlings but no difference in mature mutant seeds were not due to the different lipid extraction methods.

2) This study demonstrated that the breakdown of storage lipids in Arabidopsis (an oilseed species) is not required to supply the energy for hypocotyl elongation. It remains unanswered regarding the source of energy to support heterotrophic growth. No data are provided to support that seed storage proteins are the alternative source in ibr10 mutant.

3) Figure 5D is the only data to challenge the belief that sucrose content is correlated with hypocotyl length. Despite the importance of such data, only one allele (ibr10-1) was used whereas most of other data in this manuscript included two different alleles (ibr10-1 and ibr10-2).

Other comments:

Line 362, “quantitative metabolic approach” should be “quantitative metabolomic approach”.

---

## [Reviewer Report]

*Comments to Author*: Dear Dr. Ferjani

Thank you for submitting to QPB. We have now received reports from two reviewers. On the basis of their comments, we have decided to invite an additional revision of your work.

Below is my understanding of this study:

Elongation of hypocotyl of etiolated seedlings requires both carbon and energy sources and mobilization of TAG in seeds can serve both purposes (carbon and energy). Class II compensation, by reducing cell numbers with cell size increase, can reduce the consumption of carbon and energy required the hypocotyl elongation. In this study, the authors compared the metabolic changes of ibr10 mutant (with Class II compensation) with that of ibr1 and ibr3 mutants (no Class II compensation) (Fig. 1). The authors further showed that mobilization of TAG into sucrose is hampered in the ibr10 mutant. By carrying out BL-SOM analysis, the authors identified that Thr content correlates with hypocotyl length and showed that addition of exogenous Thr can promoter hypocotyl of all lines including WT (Fig. 5A). Then the authors suggested that Thr is a potential regulator of hypocotyl elongation (line 316). Based on the normal hypocotyl phenotype of ibr10 mutant, the authors also pointed out that Suc levels are not always correlated with etiolated seedling length (abstract).

As pointed out by the reviewers, a clearly picture of the roles of sucrose and Thr in hypocotyl elongation and the role of IBR10 should be further discussed and supported by additional data or literature. In addition to the referee reports, I have some further questions on the findings of this study.

1. What are the levels of sucrose and Thr (and other amino acids) in ibr1 and ibr3 mutants? Can the author provide a table to compare the levels of these metabolites (sugar, amino acids, with statistics) in all 3 ibr1/3/10 mutants with that of WT?

2. Is the ability of TAG mobilization different between the ibr1/3/10 mutants? These two questions important because the lower sucrose level in ibr10 mutant can be due to both 1. Lower supply of sucrose from TAG degradation, AND 2. Higher efficiency in consuming sucrose in ibr10 mutant for hypocotyl elongation, AND 3. Different strategy in resource allocation (see point 3 below).

3. Are the chlorophyll contents (SPAD values) different between WT and ibr1/3/10 mutants? It is also possible that ibr10 mutant can generate longer hypocotyl than the other mutants because it diverts its energy from chlorophyll synthesis and chloroplast biogenesis to hypocotyl elongation. TEM photos (Like Fig. 4C), if available, can also be provided.

4. As pointed out by one reviewer, why threonine is involved in hypocotyl elongation? Can Thr act as an alternative energy source or carbon source for hypocotyl elongation? Please discuss.

5. As pointed out by one reviewer, other amino acids should be tested as controls to see if Thr is unique or the same effects can be achieved by some other amino acids. At least this should be done on WT. Seedling photos and bar chart with statistics should be provided. In Fig. 5E, are the differences (+/- Thr) statistically significant? Quantitative analyses are important for QPB articles.

I hope you agree that answering the questions above and in the referees’ reports can help you to further clarify the importance of Thr and sucrose in hypocotyl elongation and why ibr10 mutant is so different from ibr1/3 mutants. Please kindly provide a letter with point-by-point responses to our comments/questions in your revised manuscript.

Yours sincerely

Boon Leong LIM

Associate Editor

---

## [Reviewer Report]

September 1, 2022

Editor-in-Chief,

Dear Prof. Olivier Hamant,

We would like to thank you very much and the two reviewers for your highly positive evaluation of our manuscript QPB-22-0013 entitled "Skotomorphogenesis exploits threonine to promote hypocotyl elongation" that you have found very interesting, and will be considered appropriate for publication in Quantitative Plant Biology if we return an acceptably revised version that is responsive to your and the reviewers comments and suggestions.

The manuscript has been revised extensively and carefully according to your advices and the comments of the Associate Editor, and the reviewers. First, we are pleased to inform you that all major areas of concern as mentioned by the Associate Editor, reviewers 1 and 2 have been properly addressed either by providing additional experimental data, or through appropriate additions and modifications to the manuscript text.

More specifically, we measured again the SPAD values in all genotypes (Col-0, ibr1, ibr3, and both ibr10 alleles), provided a new data set of the effect of exogenous supply of amino acids other than threonine (Gly, Ser, Cys, Lys, Glu, Asp, and Phe), and another data set showing the hypocotyls of all genotypes shown in Figure 5a, but this time in the presence of sucrose. All these new data sets were integrated into the revised MS as Figure 4b, Supplementary Figures S4 and S7. Despite the large amount of new data and their related description in Figures, the results and discussion, the final manuscript size did not significantly change. Therefore, we believe that we have properly generated a well-condensed version that properly addresses the reviewer’s concerns.

Here below, please find our point-by-point responses to the specific and minor comments raised by the Associate Editor, and both reviewers. As you may have appreciated, we could reach a substantially well-revised version of the manuscript that is fully responsive to your concerns and criticisms. We believe that the revised manuscript will satisfy you, the Associate Editor and the reviewers and that is much better suited for publication in QPB.

I look forward to hearing from you.

Sincerely,

Ali FERJANI

Ph. D (Associate Professor)

Tokyo Gakugei University

---

## [Reviewer Report]

*Comments to Author*: The method of lipidomic analysis for etiolated seedlings is still missing in the revised manuscript.

---

## [Reviewer Report]

*Comments to Author*: Dear Authors

Thank you for resubmitting your revised manuscript to QPB. The revised version has added additional experimentment data and most of the reviewers’ comments were answered.

As shown in the Supplementary Fig. S7, addition of Thr, but not of Gly, Ser, Cys, Lys, Glu, Asp, and Phe did not enhance the growth of etiolated seedlings. You also cited that’. As Ala, Cys, Gly, Trp, and Thr from protein reserves represent a major source of pyruvate (Eastmond et al., 2015)” (Line 367). Then, you suggested that “our data indicate that Thr can act both as an alternative energy source and a carbon source during hypocotyl elongation.” (Line 370). However, your data showed that Cys and Gly did not enhance hypocotyl elongation like Thr did. Hence, the statement in line 370 is ambiguous. There are also other possibilities. For example, Thr, rather than the a.a. you tested, can also be used for succinyl-CoA production. In the author response you wrote “Yet, at the moment we do not know the reason behind this difference among the assayed amino acids.”. Perhaps you should admit that further studies are required for finding the mechanism.

Minor points:

1. The method of lipidomic analysis for etiolated seedlings is still missing in the revised manuscript.

2. In page. 25, Supplemental Figure 6 is missing. Sometimes Figure x was used, and sometimes Figure Sx was used. Please amend.

Yours sincerely

Boon Leong Lim

---

## [Reviewer Report]

*Comments to Author*: It was pointed out in the last revision that the method of lipidomic analysis has been provided for mature seeds but not etiolated seedlings. Here, the authors have revised it to declare that the same method was applied to both tissue types. I have one last concern about the amount of specimens for analysis as Line 436 now states that approximately 0.8 mg of mature seeds or etiolated seedlings were used. However, it is suspected that one seedling already weighs >0.8 mg. It is more logical to expect that more seedlings were pooled to minimize variations in this experiment. May I ask the authors to double check before this manuscript is sent for production after acceptance? Thank you!

---

## [Reviewer Report]

*Comments to Author*: Dear Authors

Here is the comment from a reviewer. Please double check the seed weight and amend it if necessary.

“It was pointed out in the last revision that the method of lipidomic analysis has been provided for mature seeds but not etiolated seedlings. Here, the authors have revised it to declare that the same method was applied to both tissue types. I have one last concern about the amount of specimens for analysis as Line 436 now states that approximately 0.8 mg of mature seeds or etiolated seedlings were used. However, it is suspected that one seedling already weighs >0.8 mg. It is more logical to expect that more seedlings were pooled to minimize variations in this experiment. May I ask the authors to double check before this manuscript is sent for production after acceptance?”

Thank you!

Yours sincerely

Boon Leong LIM

https://boon-leong-lim-lab.webflow.io/

---

## [Reviewer Report]

*Comments to Author*: Dear Authors

We are pleased to accept your manuscript.

Regards

Boon Leong Lim